# Direct Pre-lithiation of Electropolymerized Carbon Nanotubes for Enhanced Cycling Performance of Flexible Li-Ion Micro-Batteries

**DOI:** 10.3390/polym12020406

**Published:** 2020-02-11

**Authors:** Vinsensia Ade Sugiawati, Florence Vacandio, Neta Yitzhack, Yair Ein-Eli, Thierry Djenizian

**Affiliations:** 1Mines Saint-Etienne, Center of Microelectronics in Provence, Department of Flexible Electronics, F-13541 Gardanne, France; vinsensia.sugiawati@gmail.com; 2CNRS, Electrochemistry of Materials Research Group, Aix Marseille Université, MADIREL, UMR 7246, F-13397 Marseille CEDEX 20, France; florence.vacandio@univ-amu.fr; 3Department of Materials Science and Engineering, Technion-Israel Institute of Technology, Haifa 3200003, Israel; neta_y@campus.technion.ac.il (N.Y.); eineli@technion.ac.il (Y.E.-E.); 4The Nancy & Stephan Grand Technion Energy Program, Technion-Israel Institute of Technology, Haifa 3200003, Israel; 5Al-Farabi Kazakh National University, Center of Physical-Chemical Methods of Research and Analysis, Tole bi str., 96A, Almaty 050040, Kazakhstan

**Keywords:** carbon nanotubes, polymer electrolyte, Li-ion micro-batteries, flexible anode, pre-lithiation

## Abstract

Carbon nanotubes (CNT) are used as anodes for flexible Li-ion micro-batteries. However, one of the major challenges in the growth of flexible micro-batteries with CNT as the anode is their immense capacity loss and a very low initial coulombic efficiency. In this study, we report the use of a facile direct pre-lithiation to suppress high irreversible capacity of the CNT electrodes in the first cycles. Pre-lithiated polymer-coated CNT anodes displayed good rate capabilities, studied up to 30 C and delivered high capacities of 850 mAh g^−1^ (313 μAh cm^−2^) at 1 C rate over 50 charge-discharge cycles.

## 1. Introduction

Li-ion batteries (LIBs) have been successfully employed in a wide range of applications, such as electric vehicles, microelectronic devices, etc., due to their remarkable properties such as high energy density, lack of memory effect, long cycle life, low self-discharge and high thermal resistance [1,2,3]. A large variety of carbon-based materials for LIBs have been widely investigated, such as graphene, fullerene and carbon nanotubes (CNT) [4,5,6,7,8,9,10]. For instance, a high capacity of a sandwich-like and porous NiCo_2_O_4_@reduced graphene oxide (rGO) nanocomposite serving as anode material was reported by Huang’s group [11]. Kumar et al. synthesized octahedral iron oxide nanocrystals on reduced graphene oxide nanosheets by a microwave-assisted process [12]. Zhang et al. demonstrated that electrodeposition can be used to prepare NiCo_2_O_4_/graphene [13], while Chen et al. prepared graphene hybrid nanosheet arrays via a one-pot procedure [14]; these synthesized graphene electrodes delivered good electrochemical performances. More recently, rapid research progress has been made in exploring flexible anode materials delivering high storage capacity and remarkable long-term cyclability [15,16,17]. As an allotrope of carbon, CNT electrodes offer several outstanding properties, such as excellent flexibility, fast charge transport, large surface-to-volume ratio, good chemical stability, high electrical conductivity and high reversible capacity [18,19,20,21,22,23,24,25,26]. Several methods to synthesize CNT have been demonstrated, e.g., chemical vapor deposition (CVD) [25,27], pyrolysis [28], arc discharge [29,30], laser ablation [31] and electrolysis [32]. These methods allowed the growth of CNT with various morphology, structure and properties. 

As a potential flexible anode material, our previous report showed that the CNT reached a higher specific capacity (more than 700 mAh g^−1^) compared to a traditional graphite anodes (372 mAh g^−1^) [33]. Also, Yoon et al. [34] reported that heat-treated CNT can deliver a reversible capacity of 446 mAh g^−1^ at 0.5 C, with a low Initial Coulombic Efficiency (ICE) of 9.6%. Zhou et al. [35] reported that CNT showing bamboo-like structure delivered a reversible capacity of 135 mAh g^−1^ with ICE of 17.3%. Li et al. studied a high-concentration, nitrogen-doped CNT anode providing reversible capacity of 494 mAh g^−1^ [21], while Welna et al. [36] demonstrated that vertically aligned MWCNT-based anodes showed an excellent lithium storage capacity of 980 mAh g^−1^ in the initial cycle which stabilized after first few cycles, delivering a discharge capacity of 750 mAh g^−1^. Despite CNT exhibiting remarkable features as flexible anode materials, major challenges, such as the huge surface area of the CNT promoting a high capacity loss in the first initial cycles, needs to be first tackled [37,38]. Consequently, CNT-based electrodes have a low ICE due to the solid electrolyte interphase (SEI) film formation upon the initial lithiation. This SEI film formation consumes a high amount of Li ions during the first discharge, which further limits the electrochemical performance of the cells, particularly when the CNT anode is coupled with the cathode material in full-cell configuration [39,40,41].

Several pre-lithiation methods have been investigated to compensate the severe capacity loss of the anode materials, for instance, Seong et al. reported that SiO anode was effectively pre-lithiated using stabilized lithium metal powder (SLMP) [42], Liu et al. studied pre-lithiated Si nanowire anode via a self-discharge mechanism [43] and Scott et al. also reported a complete diminishing of the initial capacity loss for carbon electrodes using n-butyllithium in hexane [44]. Wu et al. successfully fabricated a PPy/Li_2_S/KB cathode by pre-lithiation and the lithiated cathode exhibited high capacity of 1000 mAh g^−1^ with a coulombic efficiency around 95% at 0.2 C [45]. Among these methods, the self-discharge or direct pre-lithiation presents several advantages, such as low cost, fast pre-lithiation process (less time consuming), easy operation and an excellent lithiation efficiency [39,43]. However, there have been only few attempts to reduce the high irreversible capacity of CNT as anodes [39] and no study on pre-lithiated CNT coating with a polymer film. 

Recently, our groups demonstrated the crucial impact of the electropolymerization of p-sulfonated poly(allyl phenyl ether) (SPAPE) electrolyte on the electrochemical performances of CNT anodes [33]. A high reversible capacity of 750 mAh g^−1^ (276 µAh cm^−2^) at 1 C rate with the ICE of 10.4% can be obtained and interestingly, the areal capacity of SPAPE-coated CNT is enhanced by 67% compared to the pristine ones [33]. However, due to their low ICE, in the current study we report a simple direct pre-lithiation method to alleviate the initial irreversible capacity of the electropolymerized CNT anodes for flexible micro-batteries. Indeed, the long-term and extensive cycling performance of the pre-lithiated CNT is achieved over 500 cycles at a 10 C rate and its excellent rate capability is also present even at very high current density (1 to 30 C rates).

## 2. Materials and Methods

### 2.1. Materials

CNT were provided by Tortech NanoFibers Ltd. (Ma’alot Tarshiha, Israel), having a density of 0.613 g cm^−3^, a thickness of 30 µm and a porosity of approximately 70% [3]. The samples were initially washed in iso-propyl alcohol before being used as anode material, as previously reported [19,37]. A Cu target was purchased from Neyco (VANVES, France) purity: 99.9%). Lithium bis(trifluoromethane)sulfonimide (LiTFSI), 1 M LiPF_6_ (EC: DEC, *v*/*v*) and dimethylsulfoxide (DMSO) were purchased from Sigma–Aldrich (St. Quentin Fallavier Cedex, France).

### 2.2. Synthesis of the Positive Electrodes

LiNi_0.5_Mn_1.5_O_4_ (LNMO) serving as cathode was synthesized by a sol–gel method as described in [46]. To prepare the composite cathode, LNMO powder was mixed with carbon black (Super P) and polyvinylidene fluoride (PVDF) at the ratio of 80:10:10 and ground in a mortar for 20 min. n-Methyl-2-pyrrolidone (NMP) was added in the powder mixture to obtain a paste. The paste was subsequently spread on an aluminum disk as current collector with a diameter of 8 mm. The electrode was dried at 80 °C and was kept under vacuum at 110 °C for 10 h.

### 2.3. Elctropolymerization and Direct Pre-Lithiation of CNT Electrodes

The metallic Cu thin film was deposited onto the CNT surface utilizing a Cu target by radio frequency sputtering (MP300 model, PLASSYS, Marolles en Hurepoix, France), as described in our previous work [33]. The 300 nm-thin layer of sputter-deposited Cu film was utilized as the backside connection of the electropolymerization reaction. Electropolymerization of the SPAPE polymer electrolyte onto CNT was conducted by cyclic voltammetry (CV) in a three-electrode electrochemical cell using a VersaSTAT 3 potentiostat (Princeton Applied Research, Elancourt, France) with a Pt electrode as the counter and Ag/AgCl (3 M KCl) as the reference electrode [33,47]. Cyclic voltammetry was performed onto the CNT electrodes in order to polymerize the sulfonated aromatic precursor. An electrolyte solution containing 5.2 × 10^−3^ M of the synthesized monomer was mixed with 0.5 M LiTFSI as a supporting electrolyte and DMSO as a solvent. The CV experiments were carried out at room temperature for 10 cycles at the scan rate of 20 mV s^−1^ in the potential window of −0.9 to −1.8 V vs. Ag/AgCl (3 M KCl). After electropolymerization, a simple pre-lithiation treatment was achieved by pressing a Li foil and the CNT soaked with 2 drops of electrolyte composed of 1 M LiPF_6_ (EC: DEC, *v*/*v*). Various pre-lithiation durations were investigated i.e., 1, 3, 15 and 30 min, respectively. The pre-lithiated CNT were then tested as anode in half-cell and full-cell configuration. It is important to note that the used Li foil was cleaned and reutilized.

### 2.4. Characterization and Measurements

The surface morphology of the CNT anode was examined using a field-emission scanning electron microscope (SEM, Ultra-55, Carl Zeiss, Oberkochen, Germany) and by transmission electron microscopy (TEM) (Tecnai G2, Thermofisher Scientific, Waltham, MA, US). The purity of CNT was examined by x-ray diffraction (XRD) using a Diffractometer D5000 (Siemens, Munich, Germany) with CuK_α1_ (λ= 1.5406 Å) radiation, then analyzed by comparing with the JCDS-ICDD database (Joint Committee on Powder Diffraction Standards - International Center for Diffraction Data) to check the purity of the samples. Raman spectra were recorded with XploRA Raman spectrometer (Horiba Scientific, Kyoto, Japan) equipped with a 532 nm laser. For the electrochemical performance tests, Pristine CNT and SPAPE-coated CNT having a surface area of 0.44 cm^2^ were used as electrodes without the use of any binders and conductive additives, assembling using standard two-electrode Swagelok cells. All cells assembly were conducted in a glove box, filled with high purity argon (Ar) in which the moisture and oxygen contents were less than 2 ppm. Cyclic voltammetry tests were carried out using a VMP3 (Bio Logic, Seyssinet-Pariset, France) in the potential window of 0.01–2 V vs. Li/Li^+^ with a scan rate of 0.2 mV s^−1^. Galvanostatic charge–discharge cycles were performed with a VMP3 (Bio Logic) in the potential window between 0.01 and 2 V vs. Li/Li^+^ and the current density for the CNT electrodes (pristine and pre-lithiated) were 0.12, 0.24, 0.60, 1.20, 1.8, and 3.6 mA cm^−2^, respectively. The pristine CNT–LNMO and pre-lithiated CNT-LNMO full-cells were cycled at 1C and 2C rate in the potential window of 2.5–4.4 V.

## 3. Results and Discussion

### 3.1. Structural and Morphological Characterization

As shown in Figure 1a, the Raman spectrum of the CNT exhibits the two main bands characteristic of carbon materials in general, and to CNT in particular [48,49]. The G band (1573 cm^−1^) is correlated with the stretching of the C–C bond (sp^2^), and the D band (1341 cm^−1^) is attributed to the presence of disorders in the sp^2^ structure. The ratio between these two peaks (I_D_/I_G_) is often used to assess the relative content of defects in the CNT. The low D band intensity of the CNT used in this work (I_D_/I_G_ = 0.21 ± 0.02) indicates its relative purity. The crystallinity of the CNT was verified using X-ray diffraction and the patterns are given in Figure 1b. Two distinguishable diffraction peaks are clearly seen: a strong C (002) peak at approximately 26° represents the characteristic of graphite peak and the peak at approximately 43° is attributed to the (100) planes of the nanotube structure. 

According to the SEM examinations, the pristine CNT are present as highly disoriented shaped nanotubes with diameters ranging from approximately 5 to 30 nm as shown in Figure 2a,b. After electropolymerization, densified and bundled CNT are obtained due to Van-der-Waals interactions among the neighboring tubes (Figure 3c,d). Moreover, we also observed that the electropolymerization process allows the densified and bundled CNT formation due to an interaction existing between the neighboring tubes [33]. The presence of nanoparticles attached on the pristine CNT surfaces is detected from the SEM and TEM images which is assumed to be a catalyst residue of metal impurities resulting from the synthesis process. The elemental energy dispersive x-ray spectroscopy (EDS) analysis spectra shows the presence of Fe particles which are attributed to the catalyst impurities and the Cu signal came from the Cu grid.

### 3.2. Cyclic Voltammetry

In the present study, we implemented a direct pre-lithiation method in order to suppress the irreversible capacity loss during the formation of SEI film in the first cycle [39]. A pre-lithiation process is schematically depicted in Figure 3a. The CNT were directly contacted with a Li metal film in a small amount of electrolyte. Hence, a self-Li-ion discharging process (lithiation) occurs due to the short being made; the different potential existing between CNT materials and Li metal becoming reduced to zero potential. In order to follow the effect of the pre-lithiation process at various contact times (1, 3, 15, and 30 min), we first investigated the electrochemical behavior of the pristine CNT and pre-lithiated CNT electrodes by comparing the cyclic voltammetry (CV) curves, as displayed in Figure 3b–f). It is clearly observed that prior to pre-lithiation, the CV curve exhibits an open circuit voltage (OCV) of ~3.1 V vs. Li/Li^+^ and this would correspond to a purely delithiated state. During the first cycle, an obvious cathodic peak around 0.6 V attributed to the electrolyte decomposition is easily observed and corresponds to the formation and deposition of a SEI layer [50,51]. The peak disappearance in the subsequent cycles indicates a stable as-formed SEI is obtained. Another cathodic peak close to 0 V is attributed to Li^+^ intercalation into CNT and oxidation peak located at 0.25 V corresponds to the Li^+^ extraction process [21]. The broad oxidation peak ca. 1.25 V is also visible, corresponding to the extraction of Li ions from the cavities existing in the CNT structure and the small oxidation peak ca. 1.8 V might be ascribed to the reaction of lithium with hydrogen functional groups on the CNT surfaces [33]. 

After a pre-lithiation of 1 min, the OCV drops to 2.4 V vs. Li/Li^+^, suggesting that a partial insertion of lithium ions into the CNT electrodes already occurred. We also observed the noticeable cathodic peak at ~0.6 V vs. Li/Li^+^ being still present (with a lower absolute current density compared to pristine CNT), suggesting an incomplete SEI film formation on the CNT surfaces. Nonetheless, the direct pre-lithiation on the CNT surface provides beneficial effects, as the cathodic peak intensity of the SEI film formation has been successfully diminished. The observation is continued by examination of the CV curve after pre-lithiation for a period of 3 min and indeed, the OCV value decreased from 2.4 to 1.1 V vs. Li/Li^+^. At this stage, the electrode still exhibits similarity to previous samples; namely, the cathodic peak at ~0.6 V vs. Li/Li^+^ is visible but its intensity obviously weakens. We assume that the contact time of 3 min is insufficient to fully form a stable SEI film. Therefore, it was suggested that OCV should be below the cathodic peak potential of the SEI film (approximately 0.6 V vs. Li/Li^+^). Thus, the contact time during pre-lithiation process should be longer than 3 min. To further investigate the different stages of OCV value, the CNT electrode was directly contacted with Li metal for 15 min. As seen, the OCV shifted to much lower voltage of 0.45 V vs. Li/Li^+^ and the cathodic peak at ~0.6 V disappeared. After 30 min of pre-lithiation, CV curves show a slight decrease in the OCV, down to ~0.25 V vs. Li/Li^+^ without any large cathodic peak. These results reveal that the SEI film has been successfully pre-formed on the CNT surfaces for a pre-lithiation time longer than 15 min. Compared to electrochemical pre-lithiation in which the lithiation reaction requires a slow rate (usually in 10 or 20 h), the direct pre-lithiation approach is much faster [20].

### 3.3. Galvanostatic Charge–Discharge Profiles

In order to gain a deeper insight into the effect of the pre-lithiation process at different contact times, the galvanostatic charge–discharge tests were performed in a half-cell battery configuration. The charge–discharge curves for five CNT samples recorded at a current density of 3.6 mA cm^−2^ (2 C) are illustrated in Figure 4a–e. As can be seen, the various pre-lithiation times result in different electrochemical performances when the CNT are used as flexible anode materials. The plotted data shows the first two cycles corresponding to the discharge and charge processes of the pristine CNT and pre-lithiated CNT electrodes (1, 3, 15, and 30 min). During the first discharge of the pristine CNT, the voltage drops rapidly from OCV to 0.8 V vs. Li/Li^+^ with a large discharge plateau corresponding to the SEI film formation consisting of a mixture of organic and inorganic lithium compounds [52,53]. This pristine CNT provides a charge capacity of 8401 mAh g^−1^ (3091 μAh cm^−2^) and a discharge capacity of 972 mAh g^−1^ (358 μAh cm^−2^) at 2 C rate, holding ICE value of 11.57%. The very low ICE and a vast majority of irreversible capacity loss is related to the SEI formation [54]. In agreement with the CV results, the OCV of the pre-lithiated CNT electrodes decreases along with increment contact time. 

After a short contact time of 1 min (Figure 4a), the OCV drops to ~2.4 V, which is lower than that of a pristine CNT electrode. The comparison of the first discharge capacity of 6802 mAh g^−1^ (2503 μAh cm^−2^) and the first charge capacity of 843 mAh g^−1^ (310 μAh cm^−2^) leads to an ICE of 12.38%. In good agreement with the previous CV results, the OCV of the cells utilizing a contact time of 3 min is 1.1 V vs. Li/Li^+^ with a first discharge capacity of 5518 mAh g^−1^ (2030 μAh cm^−2^) and a charge capacity of 1166 mAh g^−1^ (429 μAh cm^−2^), yielding an improved ICE of 21.13%. Then, the ICE increases significantly up to 68.30% after a 15-min pre-lithiation treatment, corresponding to the first discharge and charge capacity of 1714 mAh g^−1^ (631 μAh cm^−2^) and 1170 mAh g^−1^ (431 μAh cm^−2^), respectively. Eventually, after pre-lithiation for 30 min, the capacity of the first discharge and charge cycles are 956 mAh g^−1^ (352 μAh cm^−2^) and 1179 mAh g^−1^ (434 μAh cm^−2^), respectively, resulting in a preloaded capacity of 223 mAh g^−1^ (82 μAh cm^−2^) instead of capacity loss. Indeed, the charge–discharge potential profiles of the pre-lithiated CNT electrodes are consistent with the CV curves. These results suggest that longer pre-lithiation periods (i.e., 15 min) result in fully lithiated CNT electrodes and one should also note the remarkable enhancement in the ICE of the CNT electrodes. Herein, we highlight that the degree of pre-lithiation needs to be carefully controlled. From the CV and galvanostatic curves, 15-min pre-lithiation showed high capacity and high ICE value. Pre-lithiation for 30 min is assumed lead to over-lithiation which would result in lithium plating, short circuits, and also increase the side reactions during cycling due to the excessive Li-ions on the anode surface [55]. Additionally, pre-lithiation for 1 and 3 min are insufficient due to the observable SEI formation features. As a result, this SEI depletes the cyclable lithium from the cathode material in full-cell configuration.

### 3.4. Morphology of CNT Electrode after Direct Pre-Lithiation

SEM images show the influence of direct pre-lithiation on the CNT surface morphology. The significant change of the morphology occurred, as displayed in Figure 5. After a very short contact time (1 min pre-lithiation), the bundled and densified morphology of CNT is preserved with only a few modifications of the CNT surfaces (Figure 5a,b). Surface examination is continued for 15 min pre-lithiation (Figure 5c,d), where the CV curves and galvanostatic cycling results support a stable SEI film formation, being pre-formed on the CNT surface upon self-discharge. Interestingly, and as expected, a polymer-like film covering the CNT surface can be assigned to the growth of the SEI thin film. After a 15 min pre-lithiation process, (CH_2_OCO_2_Li)_2_, polyethylene oxide, Li_2_CO_3_, LiF, Li_2_O, etc. products are presumably accumulated over the CNT surface due to the insertion and absorption of the Li ions into the CNT structures, in addition to the electrolyte reduction [50].

### 3.5. Electrochemical Performance in Half-Cells and Full-Cells

To verify the significant role of pre-lithiation prior to cell assembling and their functionalization with the polymer coating, four CNT electrodes were charged and discharged in the range of 0.01–2 V vs. Li/Li^+^ to evaluate their cycle life. Figure 6a shows the comparison of the pristine and pre-lithiated CNT. As seen, for all samples, the CNT anodes show good cycling stability, even up to 500 cycles. For the pristine CNT samples, SPAPE-coated CNT attains higher reversible capacity of 463 mAh g^−1^ (170 μAh cm^−2^) compared to pristine CNT (242 mAh g^−1^, 89 μAh cm^−2^), while for the pre-lithiated samples, the pristine CNT and SPAPE-coated CNT yielded a reversible capacity of 356 mAh g^−1^ (131 μAh cm^−2^) and 508 mAh g^−1^ (187 μAh cm^−2^), respectively over 500 cycles at 10 C rate (Figure 4b). Thus, it is clear that coating CNT with the polymer electrolyte via electropolymerization reaction has a beneficial impact, resulting in the improvement of the cell performance [33,46,56,57]. The enhancement can be attributed to the combination of two effects: the larger electrode/electrolyte interface resulting in improved charge transport and a better penetration of the polymer electrolyte onto the carbon nanotube surfaces [33]. The fact that SPAPE can improve the kinetics of charge/discharge has been evidenced by electrochemical impedance spectroscopy in one of our previous works [47].

By comparing the 1st and 2nd discharge capacity of both pristine and pre-lithiated samples, the importance of direct pre-lithiation prior to cell assembling can be demonstrated (Figure 6a,b). For the non-pristine CNT, the 1st and 2nd discharge capacity of the pristine CNT are 4298 mAh g^−1^ (1581 μAh cm^−2^) and 681 mAh g^−1^ (251 μAh cm^−2^), while SPAPE-coated CNT provides 1st discharge capacity of 6699 mAh g^−1^ (2465 μAh cm^−2^) and 2nd discharge capacity of 955 mAh g^−1^ (351 μAh cm^−2^). Both pristine samples exhibited very high initial capacity loss. This irreversible capacity is considered as a critical issue, notably when assembling a full-cell configuration, since lithium is irreversibly consumed after the first lithiation. In agreement with previous findings [45,54,58], the high capacity loss can be significantly reduced after pre-lithiation. For the pre-lithiated CNT, the 1st and 2nd discharge capacity of the pristine CNT are 286 mAh g^−1^ (97 μAh cm^−2^) and 861 mAh g^−1^ (317 μAh cm^−2^), while SPAPE-coated CNT gives 1st discharge capacity of 981 mAh g^−1^ (361 μAh cm^−2^) and 2nd discharge capacity of 1188 mAh g^−1^ (437 μAh cm^−2^). Figure 6c–f show the charge and discharge profiles of the CNT anodes under four fabrication conditions: pristine CNT, SPAPE-coated CNT, pre-lithiated pristine CNT and pre-lithiated SPAPE-coated CNT, respectively. The charge and discharge profiles for all samples show pronounced sloping curves, with a cell voltage of ~0.5 V vs. Li/Li^+^. Moreover, the overlapping of the galvanostatic curves suggests a good electrochemical reversibility of the CNT anode, particularly after direct pre-lithiation.

To further examine the rate capability performance of the pre-lithiated CNT electrodes, galvanostatic charge–discharge cycling was carried out at progressively increased current density, ranging from 0.12 to 3.6 mA cm^−2^. Figure 7a,b shows the galvanostatic charge–discharge profiles of the pre-lithiated CNT with polymer coating at a potential range of 0.01–2 V at 1 C for 50 cycles. After 50th cycle the coulombic efficiency reaches approximately 93%, corresponding to a charge capacity of 792 mAh g^−1^ (291 μAh cm^−2^) and the discharge capacity is 850 mAh g^−1^ (313 μAh cm^−2^) at 1 C. Their reversible capacity is >2-fold higher compared to the storage capacity of graphite (372 mAh g^−1^). We note that capacity fading occurs in the few initial cycles which could possibly due to the defects on CNT, impurities and irreversible lithium loss due to the side reactions upon cycling. However, the capacity is stabilized after 10 cycles and compared to pristine CNT which showed a severe capacity decay (~10 times) [33], the pre-lithiated CNTs show better cycling performance.

Remarkably, even at stepwise accelerated rate, the pre-lithiated CNT could still provide excellent capacities of 764 mAh g^−1^ (281 μAh cm^−2^) at 2 C, 647 mAh g^−1^ (238 μAh cm^−2^) at 5 C, 545 mAh g^−1^ (200 μAh cm^−2^) at 10 C, 484 mAh g^−1^ (178 μAh cm^−2^) at 15 C and 385 mAh g^−1^ (142 μAh cm^−2^) at 30 C (Figure 7c). Indeed, when the current density turned back from high to low current densities, the capacity can be greatly recovered over 120 cycles. Herein, we demonstrated that a simple direct pre-lithiation can improve the electrochemical performance of the flexible CNT anode by diminishing their extensive and large irreversible capacity, as well as providing a lithium supply to compensate the lithium loss during cycling. As a result, the CNT electrode shows good cycling stability over 500 cycles and excellent rate capabilities, even at high C-rates.

The promising performances of the CNT suggested their use as potential flexible anode material for application in full lithium-ion batteries using high-voltage cathode material, i.e., LNMO, in order to achieve high energy density. The galvanostatic charge–discharge profiles of the full-cell battery using both pristine CNT and pre-lithiated CNT as anodes are presented in the Figure 8. The battery was cycled at 1 C in the potential window of 2.5–4.4 V. Since the mass limitation is controlled by CNT, the capacity is reported versus the anode material and the mass was calculated considering a porosity of 70%. In the first cycle, the pristine CNT cell exhibits an initial charge capacity of 1856 μAh cm^−2^ and a discharge capacity of 76 μAh cm^−2^ with a relatively low coulombic efficiency of ca. 4.1% (Figure 8a), ascribed to the abovementioned SEI formation at the initial charging. This SEI film reduced the amount of active lithium ion upon cycling.

In contrast, as seen in Figure 8b, the pre-lithiated CNT cell gives a superior electrochemical performance, delivering a discharge capacity of 238 μAh cm^−2^ with a cell voltage of about 3.75 V at 1 C. Another crucial point is that the OCV for the pristine CNT starts from ca. 0 V, while the OCV of the pre-lithiated CNT is ca. 4.2 V which means the SEI layer has been pre-formed on the CNT surface. For the pristine CNT, the capacity decreases 42% after 10th cycle at 1 C. The reason for this is likely ascribed to the relatively low coulombic efficiency of the pristine CNT in the half-cell (11.57%). At 2 C (Figure 8c), the pre-lithiated CNT reaches capacity of 223 μAh cm^−2^. The important differences between pristine CNT and pre-lithiated CNT became obvious after 10 cycles charge–discharge cycling as seen in Figure 8d. In the full-cell configuration, the capacity of pre-lithiated CNT is 4 times higher compared to pristine CNT.

## 4. Conclusions

In summary, the electropolymerization of SPAPE polymer electrolyte into carbon nanotubes has been conducted by cyclic voltammetry. The enhanced electrochemical performance of SPAPE-coated CNT compared to pristine ones, due to the high electrode/electrolyte interface area leads to the improved charge transfer. In addition, this study also clearly showed the positive effect of direct pre-lithiation to suppress the initial irreversible capacity of CNT. Light-weight free-standing and flexible CNT without any binder and conductive additives have been successfully utilized as anode materials in Li-ion batteries, demonstrating both a stable and a high reversible capacity of 508 mAh g^−1^ (187 μAh cm^−2^) at a 10 C rate over 500 cycles. Pre-lithiation of CNT via a self-discharge mechanism improves the first cycle coulombic efficiency from 11.57% to 68.30% after 15 min pre-lithiation period and reaches >100% subsequent to 30 min of pre-lithiation period. Moreover, by coupling the CNT anode with a high voltage LNMO spinel cathode, a 3.75 V full-cell presented a high capacity of 238 μAh cm^−2^ at a 1 C rate with the coulombic efficiency of ca. 90% in the initial cycle. This simple, fast and inexpensive method enables the achievement of high capacity flexible anodes for micro-batteries.

## Figures and Tables

**Figure 1 polymers-12-00406-f001:**
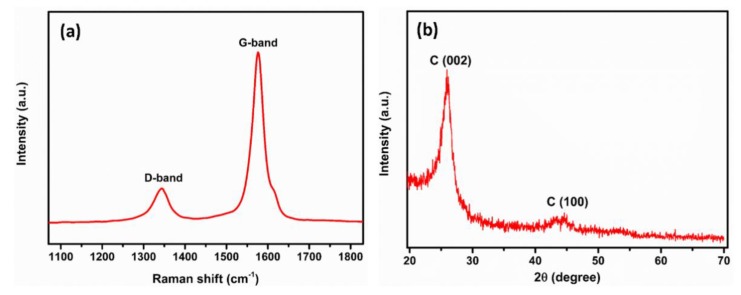
Raman spectra (**a**) and XRD pattern of the pristine carbon nanotubes (CNT) (**b**).

**Figure 2 polymers-12-00406-f002:**
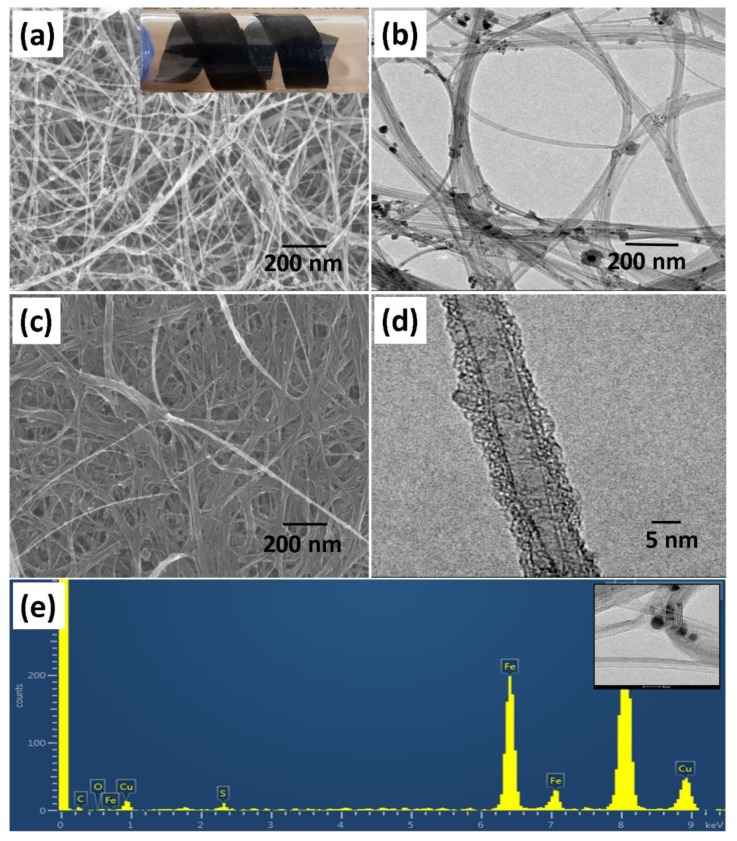
SEM image of pristine CNT (photograph of a flexible CNT in inset) (**a**); TEM image of pristine CNT (**b**); SEM image of p-sulfonated poly(allyl phenyl ether) (SPAPE)-coated CNT and (**c**) TEM image of SPAPE-coated CNT (**d**); and EDS spectra of pristine CNT (the inset is the corresponding TEM image) (**e**).

**Figure 3 polymers-12-00406-f003:**
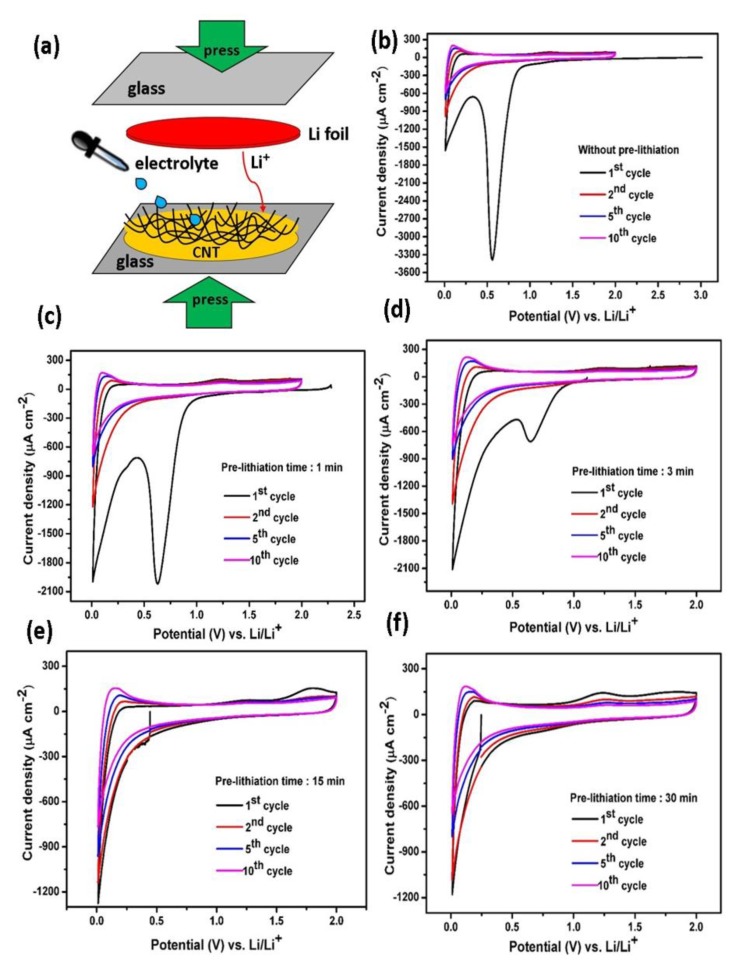
A schematic representation of a direct pre-lithiation method (**a**), cyclic voltammogram curves of the pristine CNT (**b**), pre-lithiated CNT for 1 min (**c**), pre-lithiated CNT for 3 min (**d**), pre-lithiated CNT for 15 min (**e**) and pre-lithiated CNT for 30 min (**f**) recorded at a scan rate of 0.2 mV s^−1^ in a potential range of 0.01–2 V vs. Li/Li^+^.

**Figure 4 polymers-12-00406-f004:**
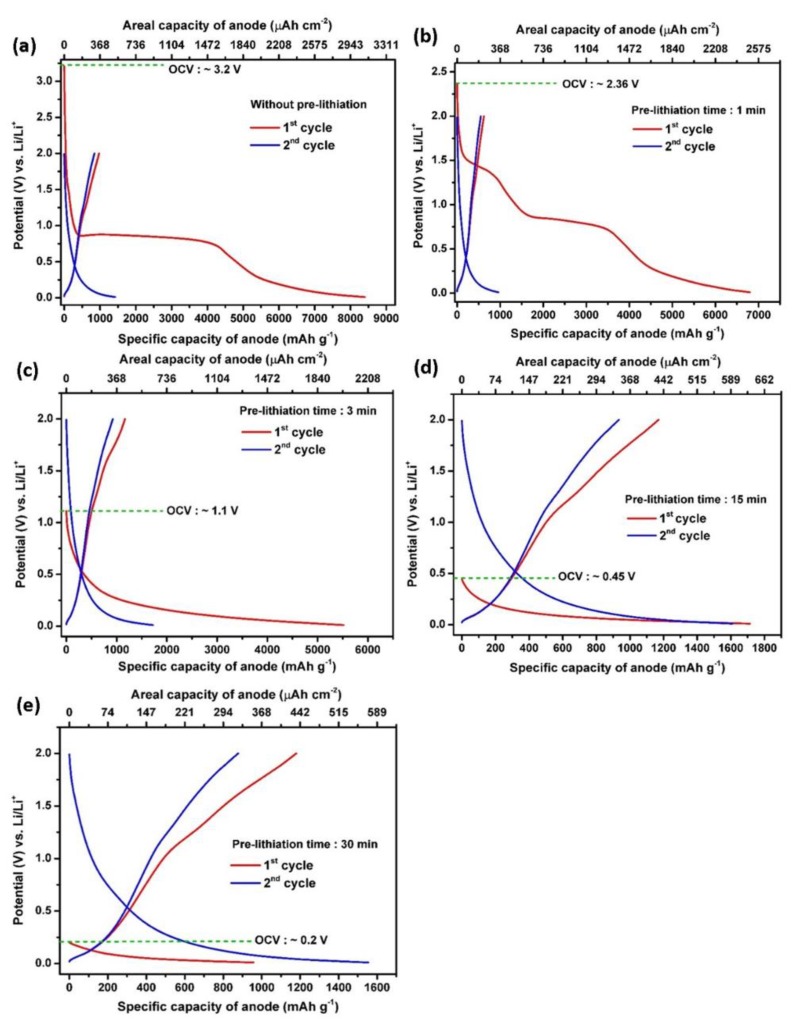
Initial charge–discharge potential profiles of the pristine CNT (**a**), pre-lithiated CNT for 1 min (**b**), pre-lithiated CNT for 3 min (**c**), pre-lithiated CNT for 15 min (**d**) and pre-lithiated CNT for 30 min (**e**) recorded at 2 C rate.

**Figure 5 polymers-12-00406-f005:**
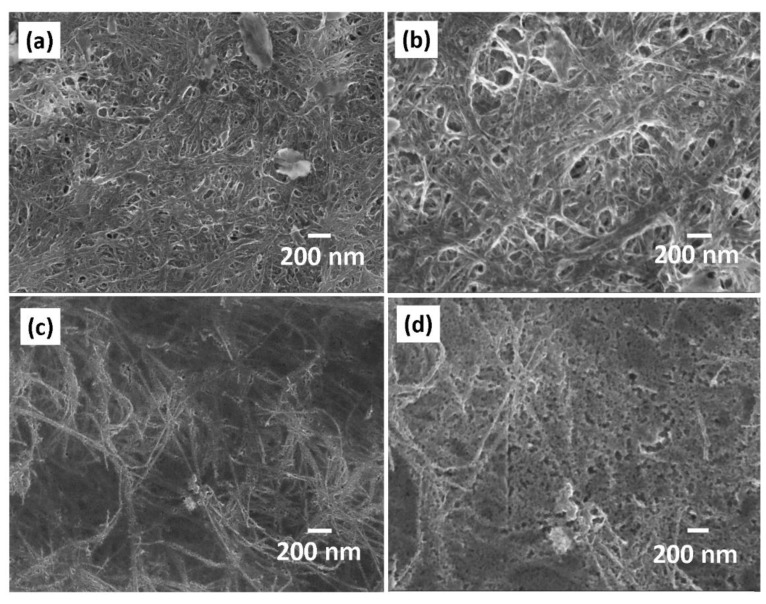
SEM images of a pre-lithiated CNT after 1 min (**a**,**b**) and pre-lithiated CNT after 15 min (**c**,**d**).

**Figure 6 polymers-12-00406-f006:**
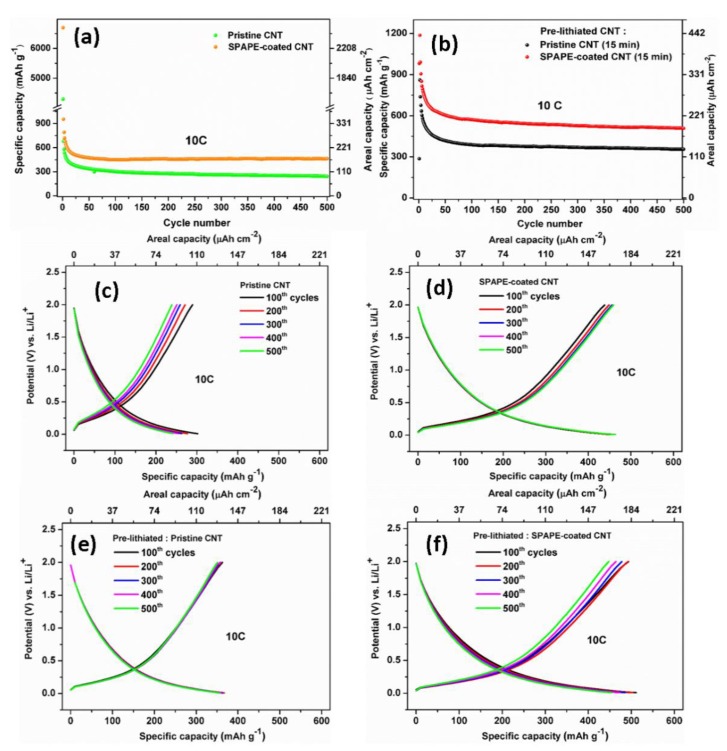
Long-term cycling performance of the (**a**) pristine and (**b**) pre-lithiated CNT anodes; galvanostatic charge–discharge curves of the (**c**) pristine CNT, (**d**) SPAPE-coated CNT, (**e**) pre-lithiated pristine CNT and (**f**) pre-lithiated SPAPE-coated CNT in the potential window of 0.01–2 V at a constant current density of 1.2 mA cm^−2^.

**Figure 7 polymers-12-00406-f007:**
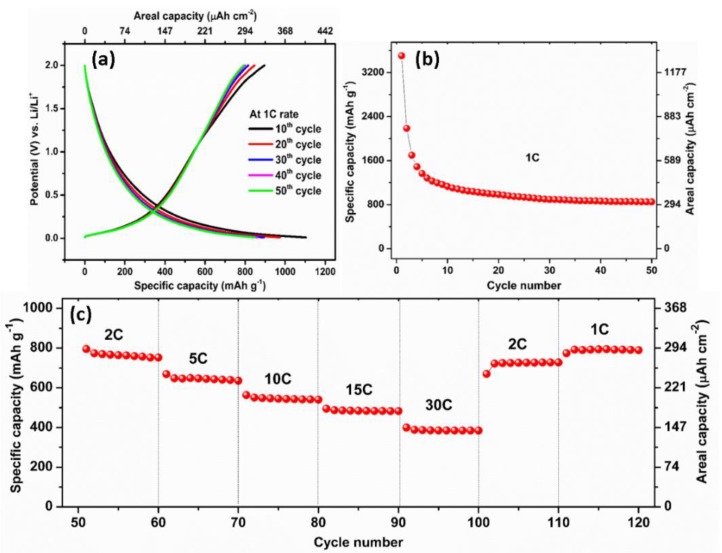
Typical galvanostatic charge–discharge potential profiles vs. Li/Li^+^ for the 10th, 20th, 30th, 40th, and 50th cycle against the capacity of the pre-lithiated CNT electrode for 15 min at 1 C rate (**a**), rate capabilities of the pre-lithiated CNT electrode for 15 min at 1 C rate (**b**) and multiple C-rates (**c**).

**Figure 8 polymers-12-00406-f008:**
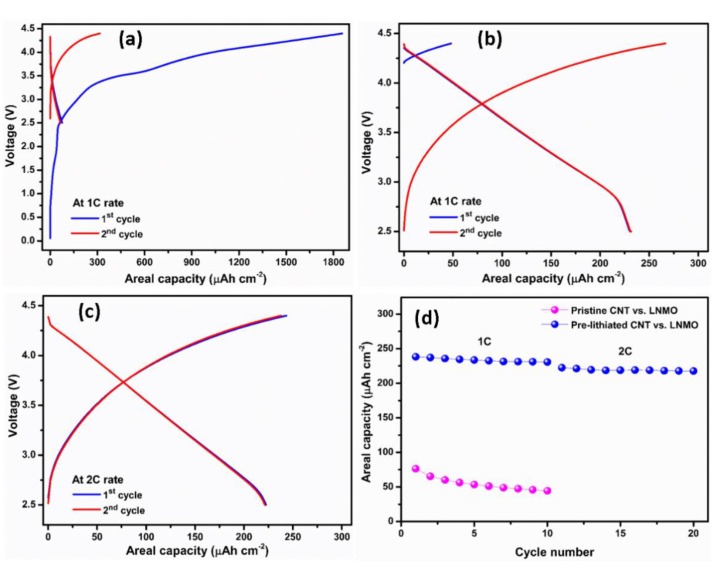
Typical galvanostatic charge–discharge potential profiles for the 1st and 2nd cycle at 1 C rate for (**a**) pristine CNT–LNMO, (**b**) pre-lithiated CNT-LNMO, (**c**) galvanostatic charge–discharge profiles of pre-lithiated CNT–LNMO at 2 C rate, (**d**) cycling performance of the pristine and pre-lithiated CNT anodes in the full-cell configuration.

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
