# Peer review of "Direct Pre-lithiation of Electropolymerized Carbon Nanotubes for Enhanced Cycling Performance of Flexible Li-Ion Micro-Batteries"

_polymers, 2020, doi:10.3390/polym12020406_

Round 1
Reviewer 1 Report
The article “Direct Pre-lithiation of Electropolymerized Carbon Nanotube Tissues for Enhanced Cycling Performance of Flexible Li-ion Micro-batteries” submitted by Sugiawati et al. is interesting as well as well written for readers. I recommend it acceptance after minor revision because several aspects are not clear at this stage so need rigorous revision. Comments are suggested below
Differentiate this research work as comparison to other research work published on similar kind of materials. What is the meaning of CNTs tissue? “Tissue”??? Explain that why tissue attached with CNTs word? What is the purpose of addition “Tissue word with “CNTs”?? Remove the tissue word from CNTs and write simple as “CNTs”. It makes strange meaning…… In abstracted “….displayed impressive rate capabilities,…” It is not looks impressive so remove the adjective “Impressive” and write in simple word. Introduction section contains lack of proper relevant references. So improve this section by recent published article as new carbon derivatives (as CNTs and graphene) for energy storage application (Battery and Supercapacitor). Articles are suggested here as Progress in Energy and Combustion Science 75, 100786, 2019; Electrochimica Acta 303, 246-256, 2019; Electrochimica Acta 281, 78-87, 2018; Journal of Alloys and Compounds 695, 1793-1801, 2017; RSC Advances 6, 52945 – 52949, 2016; Journal of Nanoparticle Research 17:24, 2015; International Journal of Nanoscience 10, 809-813, 2011; Journal of Nanoparticle Research 17:24, 2015 In Figure 3, diameter of CNTs (5-30 nm) is not clear, provide the higher magnification images using TEM. Also the SEM image (Fig 3a) contains particles witch are attached on the surface of CNTs (see carefully), What kind of these particle are available? These are catalyst (during CNTs synthesis) or contamination like impurities? Confirm it by EDS (elemental contents) and characterized by XRD to determine the structural analysis. Why SPAPE-coated CNT is improves the areal capacity compared to the pristine CNTs? What kind of mechanism occurs the improvement in application? The reason of redox peaks in Fig. 1, are not provided in the written article. Provide the reason of oxidation and reduction as appeared in CV curves. Pre-lithiation for 3 min and 30 min shows large variation in area for 1st to 10th What is the reason that longer lithiation reveals nearly equal area of CV curve? How lithiation time related with electrochemical performance for improvement the application? In Fig. 5b, why specific capacity drop very fast in initial few cycles (< 10 cycles)? It drop nearly 4 times as compared to initial value. Provide the reason. Improve the conclusion section because some of contents are common in abstract as well as in conclusion.Author Response
Please see the attachment

Reviewer 2 Report
Present work is on the effect of pre-lithiation of carbon nanotube tissue anode.
It shows good results and should be suitable for publication in Poymer.
There is one thing that should be added:
The reason why 15 min sample performed best must be discussed in detail.
Since this is a research paper, analysis of such results is important.
Reviewer 3 Report
This manuscript demonstrated the flexible carbon nanotube (CNT) tissue anode and increase of the initial coloumbic efficiency of CNT tissue by simple direct pre-lithiation. The electrochemical performance of this CNT tissue is quite good. However, before publication, the following points need to be addressed.
The CNT tissue is known as having high electrical conductivity and is applied to the current collector of the electrode. Why was the Cu sputter deposition conducted on the CNT tissue? The characteristics of the CNT tissues should be provided, such as Raman, XRD data, etc. Why the SPAPE-coated CNT showed better performance than the pristine CNT? The effect of SPAPE coating should be discussed in this manuscript. The CNT tissues showed the potential for an electrode of high power batteries. However, this manuscript showed only half-cell data. To apply the CNT tissues as an anode of high power batteries, the author needs to show that the CNT tissue electrode works well in full cell.
Round 2
Reviewer 3 Report
The authors have addressed most of the reviewer’s comments, but the manuscript still needs to improve.
The authors claimed that the SPAPE improves charge transfer during the charge/discharge. To support this discussion, AC impedance analysis is needed.Author Response
PLease see the attachment
